

# An R package for survival-based gene set enrichment analysis

Xiaoxu Deng[1] and Jeffrey Thompson[1,2]

[1] Department of Biostatistics & Data Science, University of Kansas Medical Center, Kansas City, KS, United States of America
[2] University of Kansas Cancer Center, Kansas City, KS, United States of America

## ABSTRACT

Functional enrichment analysis is usually used to assess the effects of experimental differences. However, researchers sometimes want to understand the relationship between transcriptomic variation and health outcomes like survival. Therefore, we suggest the use of Survival-based Gene Set Enrichment Analysis (SGSEA) to help determine biological functions associated with a disease's survival. Despite the availability of this method to researchers, there are no standard tools or software to perform this analysis. We developed an R package and Shiny app called SGSEA and presented a study of kidney renal clear cell carcinoma (KIRC) to demonstrate the approach. In Gene Set Enrichment Analysis (GSEA), the log-fold change in expression between treatments is used to rank genes, to determine if a biological function has a non-random distribution of altered gene expression. SGSEA is a variation of GSEA using the hazard ratio instead of a log fold change. Our study shows that pathways enriched with genes whose increased transcription is associated with mortality (NES > 0, adjusted $p$-value < 0.15) have previously been linked to KIRC survival, helping to demonstrate the value of this approach. This method allows rapid identification of disease variant pathways and provides supplementary information to standard GSEA, all within a single R package at https://github.com/ShellsheDeng/SGSEA or via the convenient app at https://biostats-shinyr.kumc.edu/SGSEA/.

## INTRODUCTION

Portions of this text were previously published as part of a preprint (https://doi.org/10.21203/rs.3.rs-3367968/v1). Functional enrichment analysis of gene expression data is a standard step following most transcriptome profiling experiments in biomedical research. It enables researchers to characterize the biological functions affected by the experiment based on a subset of relevant genes and study the drug and disease mechanisms at the molecular level (*Reimand et al., 2019*). Gene sets over-representation analysis (ORA) and Gene Set Enrichment Analysis (GSEA) are two widely used tools to perform functional enrichment analysis (*Backes et al., 2007*; *Zhao & Rhee, 2023*; *Huang da, Sherman & Lempicki, 2009*). ORA tests whether a pre-selected gene set is more highly represented in a list of genes that show significant changes in expression compared to what would be expected by chance

Corresponding author
Jeffrey Thompson,
jthompson21@kumc.edu

based on one or more statistical tests, such as the hypergeometric test (*Draghici et al., 2003*). However, the significance threshold, or the process of selecting differentially expressed genes is often arbitrary, and important genes with smaller changes (at the boundary) might be missed (*Khatri, Sirota & Butte, 2012*). To address this issue, GSEA has been proposed (*Subramanian et al., 2005*). It overcomes this by analyzing gene expression changes across all genes in a ranked list, identifying pathways that show consistent changes at the top or bottom of this list, regardless of the absolute changes in gene across any two phenotypes (*Subramanian et al., 2007*). Both methodologies rely on gene annotation databases, in which a great deal of work has been devoted to organizing genes and proteins into biological functions and pathways. Examples include Reactome, the Kyoto Encyclopedia of Genes and Genomes (KEGG), the Gene Ontology (GO) and the Molecular Signatures Database (MSigDB). KEGG was developed in 1995 (*Ogata et al., 1999*), and GO (*Ashburner et al., 2000*) was founded in 1998, between which year the first eukaryotic genomes were released to the public. Reactome (*Joshi-Tope et al., 2005*) was created in 2004, while MSigDB (*Subramanian et al., 2005*) was introduced in 2005 as a resource for GSEA. Each of these gene annotation databases has certain advantages for particular studies, however, any of them can be used to perform functional enrichment analysis.

Although functional enrichment analysis is typically focused on determining the impact of experimental differences, sometimes researchers are interested in how transcriptomic variation is associated with health outcomes, such as survival (*Nagy, Munkacsy & Gyorffy, 2021*; *Dwivedi et al., 2022*). Additionally, there is often an interest in which biological functions are associated with the outcome, rather than individual gene (*Moon et al., 2019*). Even if a specific gene is the cause of poor disease outcomes, an intervention might be possible at multiple points in the functional pathway involving that specific gene. Alternatively, there may be small cumulative impacts on a biological pathway that result in poor outcomes. However, typical functional enrichment analyses are focused on identifying biological functions associated with experimental differences, not outcomes. Although these ideas can align, the focus on disease differences can lead to an analysis missing important results. For example, in the case of cancer, some of the pathways that differ between tumor and normal samples will likely be associated with survival (*Darang et al., 2023*). However, other pathways that influence survival differences among survivors may not differ much between tumor and normal samples. That is because these differences relate to population variation that might be key to survival and these differences are not driven by the disease (*Wang et al., 2022a*). Furthermore, the analysis will not directly identify the association between pathways and survival outcomes, or its magnitude, and researchers often resort to tests of individual genes in some of their top pathways or other indirect methods. For example, several studies used GSEA to identify cancer-related pathways and then followed by survival analysis, but their methods do not directly incorporate survival information during pathway enrichment analysis, and hence their results do not reflect the degree of association between survival and biological functions (*Kim et al., 2018*; *Yang et al., 2018*). Furthermore, the survival differences observed among patients with the disease were not the primary drivers of the results in the GSEA or survival analysis, as these analyses typically

focus on treatment differences rather than survival or case differences within the disease group, leading to potential blind spots in the findings.

There have been a couple of attempts to directly link functional enrichment to clinical outcomes (*e.g.*, survival status or survival time). For example, *Woltmann et al. (2014)* identified enriched pathways through different pathway enrichment analysis tools based on a genome-wide association study (GWAS) of short-term *vs.* long-term breast cancer survival in order to make a conclusion to a more general breast cancer population globally. However, their studies mainly focused on verifying breast cancer-related pathways by showing the similarities between the GWAS method and different enrichment analyses and did not suggest an approach to handle censored survival data. Furthermore, though *Goeman et al. (2005)* developed a statistical test of the association between groups of genes and a clinical outcome based on the Cox proportional hazards model, their proposed method requires a set of pre-selected genes and does not consider coordinated changes of the genes in a pathway.

We propose a simple solution to this challenge. Researchers can perform Survival-based Gene Set Enrichment Analysis (SGSEA) to understand how transcriptomic variation among patients can be used to identify biological functions associated with survival from the disease. While GSEA is a powerful method for identifying pathways based on experimental differences, SGSEA extends this by incorporating survival data, linking gene expression patterns with disease outcomes like survival time. Traditional GSEA does not account for survival data, which is crucial for understanding disease progression. SGSEA offers a complementary approach by focusing on how gene expression impacts survival, making it particularly useful for studying diseases where survival time is a key outcome. Therefore, we present the SGSEA R package and Shiny app, which facilitate this analysis conveniently. Although researchers can already use GSEA to perform this analysis, it is not included as a standard tool in most software and there has not been a paper to demonstrate its usefulness. With these tools, researchers may uncover unexpected patterns of gene activity, leading to new avenues of research. In this paper, we will use our R package and Shiny app to demonstrate while GSEA often provides insight into disease etiology through experimental differences, SGSEA can provide additional information and unique insights into disease outcomes.

## METHODS

### Survival gene set enrichment analysis

GSEA is a powerful tool to identify the biological functions that are enriched in up- or down-regulated gene expression from comparing two treatment groups of genes. It ranks genes based on the magnitude of expression changes between two groups (*e.g.*, log-fold change), such as treatment and control, and then tests whether the genes in a particular pathway are overrepresented at the extremes of the ranked list. By examining the distribution of gene ranks in pathways, GSEA identifies biologically meaningful patterns of coordinated changes. Therefore, to perform the SGSEA, we replace the typical log-fold change (LFC) used in GSEA with log hazard ratios (LHR) to find the biological functions that are
associated with mortality or survival. The Cox model allows us to estimate how gene expression impacts the instantaneous risk of mortality, without making assumptions about the distribution of hazards over time, and it can handle censored data (cases where the exact survival time is not known). In SGSEA, we rank genes based on their log hazard ratios, which reflect the gene's association with mortality or survival. This modification allows SGSEA to identify biological functions linked to survival outcomes, which is particularly valuable for analyzing diseases like cancer, where survival is a critical endpoint.

Through this semi-parametric model, we can evaluate the change in the instantaneous risk of mortality while incorporating each individual gene's expression as the risk factor. The hazard of dying for gene $i$ at time $t$ is defined as

$$h(t,x_i) = h_0(t)e^{\beta x_i} \tag{1}$$

where $h_0(t)$ is the baseline hazard function when $gene_i$ is not expressed.

Taking the logarithm of Eq. (1) on both sides, we have

$$log[h(t,x_i)] = log[h_0(t)] + \beta x_i. \tag{2}$$

$\beta$ in Eq. (2) is the log hazard ratio for a one-unit increase in the value of $x_i$, the normalized gene expression. In summary, the SGSEA is a variation of GSEA by performing GSEA with a hazard ratio instead of a log fold change.

## R package

All analyses performed in this paper use R statistical programming language (*R Core Team, 2023*). We created the *SGSEA* R package that is designed specifically for SGSEA analysis. It provides a complete set of functions for SGSEA and standard GSEA that allow users to easily normalize gene expression data using a mean–variance model method, calculate log hazard ratios with adding optional covariates based on the Cox proportional hazard model, calculate $log_2$ fold-change based on the negative binomial regression model, extract reference pathway annotations from the Reactome database or GO terms, generate the pathway enrichment analysis statistics, export the ranked genes along with their Cox hazard ratios to a CSV file, and produce a table of both the top 10 significant pathways associated with disease survival and top 10 significant pathways associated with disease mortality as well as the individual pathway enrichment plot. Therefore, this package can also be used for the standard case *vs.* control (or normal *vs.* tumor) GSEA method, providing users with the additional advantage of conducting various functional enrichment analyses using a single package. The **coxph** function from the *survival* package is used to fit Cox Proportional Hazards model (*Therneau & Grambsch, 2023*), from which estimates of the log-hazard are obtained. Because higher hazard ratios generated by the model indicate that increased expression is associated with an increased hazard of the event, higher enrichment scores indicate that increased expression in a pathway is associated with a higher risk of mortality. We then call the **fgsea** function from the *fgsea* package in R to conduct the enrichment analysis. This package can efficiently and accurately estimate $p$-values, which significantly increases the sensitivity of GSEA in multiple hypothesis correction procedures compared to other implementations (*Korotkevich, Sukhov & Sergushichev, 2021*). In addition to our

case-only SGSEA, we also conducted the standard GSEA method between the tumor and tumor-adjacent normal tissues by using the **DESeq** function from the *DESeq2* package in R (*Love, Huber & Anders, 2014*), which uses a negative binomial generalized linear model, to get the log fold-change as the input of the enrichment analysis. Additionally, the package includes an example R script (SGSEA_example_script.R) that guides users through the SGSEA workflow by simply running R commands directly and a downloadable example data that allows users to follow easily. This package can be downloaded at the GitHub repository https://github.com/ShellsheDeng/SGSEA.git. All code and results generated by the SGSEA R package are licensed under the GPLv3 license, ensuring open-source access and usage.

## Shiny app

The SGSEA Shiny app is an interactive web application that performs SGSEA (Fig. 1). It was built using the *Shiny* package and allows users to input multiple formats of the gene expression file and corresponding survival information. The SGSEA Shiny app outputs a table of enrichment results with a series of options for data pre-processing steps such as filtering and normalization. For first-time users, the example dataset can be downloaded from the top left corner of the webpage to familiarize themselves with the expected input format and an example script is provided to guide users through the SGSEA analysis workflow in the package. The app provides a visually appealing and intuitive interface for users to interact with on the left and show results on the right. It is designed for researchers with no programming background. This app can be initiated by either calling the **runExample** function from the *SGSEA* package or by visiting the website https://biostats-shinyr.kumc.edu/SGSEA/. All code and results generated by the SGSEA Shiny app are also licensed under the GPLv3 license, covering both the software tool and the results produced.

## Data summary

To demonstrate the *SGSEA* R package and app, we used an mRNA-seq dataset that contains patients with kidney renal clear cell carcinoma (KIRC) (version 2016_01_28) from The Cancer Genome Atlas (TCGA) (*Tomczak, Czerwińska & Wiznerowicz, 2015*). Data were obtained from The Broad Institute's Genome Data Analysis Center (GDAC) Firehose website (https://gdac.broadinstitute.org/). In this dataset, there are 520 observations, out of which 360 patients were alive at the last follow-up and 160 patients were dead. After filtering out some invalid data input (*e.g.*, survival time < 0) and genes without HUGO symbols, there were 20,502 genes remaining from the original 60,660. The standard GSEA method conducted in this paper uses 70 out of those 520 patients who have paired normal tissues that allow us to conduct a case-control analysis to derive log fold-change of differential expression between tumor and normal.

We used the human pathway annotations from both the Reactome pathway database and Gene Ontology database. We mapped Entrez IDs to HUGO gene symbols using the org.HS.eg.db R package (*Carlson, 2023*). Reactome is a curated and peer-reviewed pathway database that provides detailed information on molecular processes, such as metabolism,

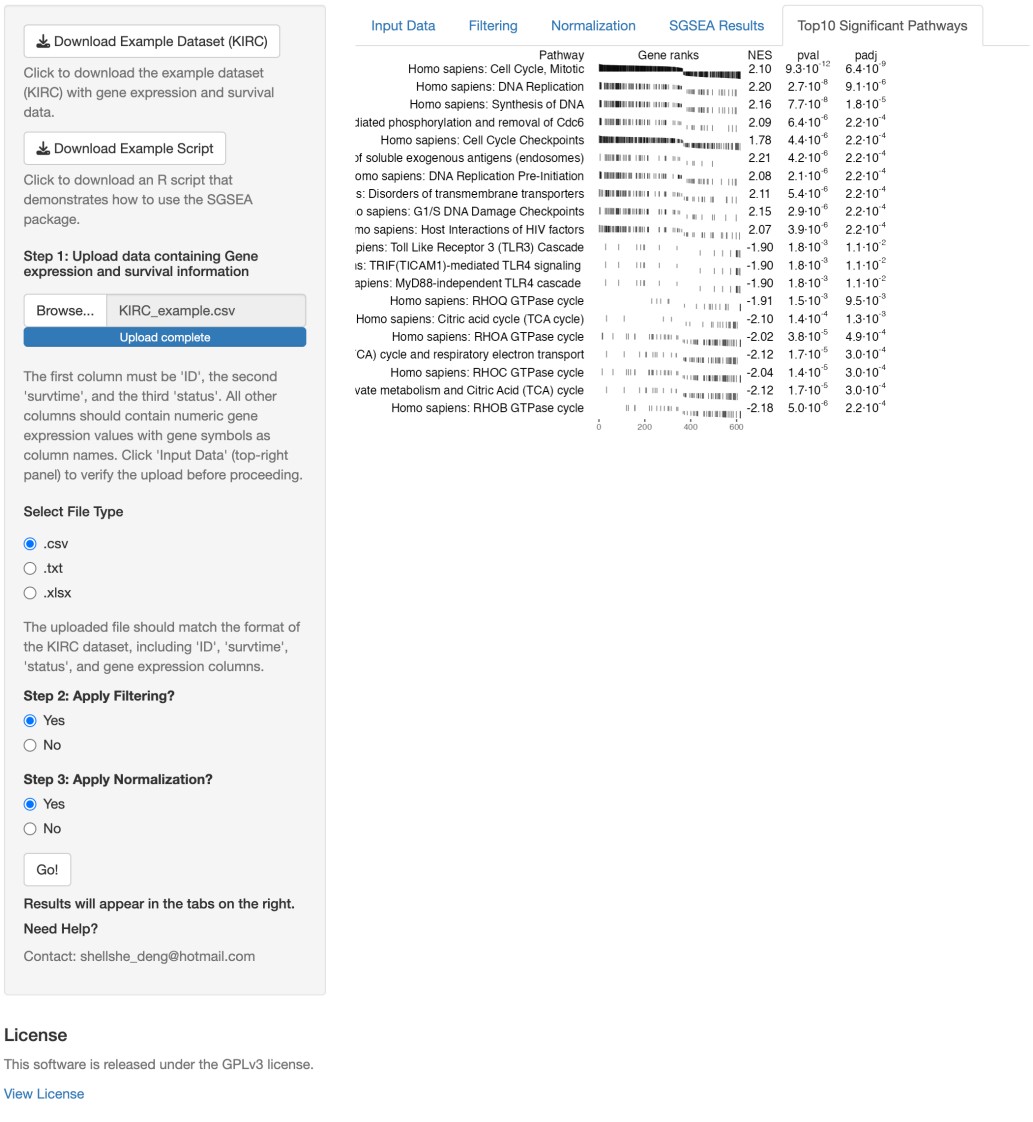

**Figure 1** **SGSEA Shiny app interface.** Overview of the Shiny app interface for conducting SGSEA analysis, showing the interactive features for pathway enrichment analysis.

signaling, and disease pathways for over 2,000 different species (*Fabregat et al., 2018*). Due to the high-quality annotations, comprehensive coverage of human biological pathways, and the most prevalent annotation systems in genomics, we selected both Reactome and GO terms as the knowledge base in this analysis to demonstrate how SGSEA can help researchers gain insights into underlying biological processes and mechanisms in their data.

## Data structure

The general data structure for conducting the SGSEA method is the same for both the R package and the Shiny app. The rows should be the patients or samples and the columns

should be the gene expression count data. To perform the survival analysis, there are two important variables that need to be included in the data, the patient's time-to-event and survival status. Users can also check the input data structure by referring to an example data called KIRC which is a subset of the original KIRC RNA-seq data that is embedded in the package as well as a downloadable example dataset function and button in both package and app.

## Data preprocessing
### Filtering

The R package users can choose different filtering methods based on their own preferences. However, the filtering rule is fixed in the Shiny app and the results presented in this paper all use the same filtering rule. Specifically, we excluded genes with less than 1 read per 10 participants on average before conducting the SGSEA method. For the KIRC data set, there were 19,098 genes that withstood our filtering procedure. When performing the typical GSEA method, this filtering rule left the total number of genes at 18,198. This difference is due to the fact that the analyses were conducted on different sets of samples.

### Normalization

In order to adjust for the heteroscedasticity and nonnormality of the gene expression data, we use the **voom** function from *limma* package (*Ritchie et al., 2015*) for both the *SGSEA* package and the Shiny app before performing the SGSEA method. This function estimates the mean–variance trend for log counts and then assigns a weight to each observation based on its predicted variance. However, the normalization method for the preprocessing step before performing typical GSEA in the *SGSEA* package is different from the one in the preprocessing step of the SGSEA method, because it calls out the **DESeq** function from *DESeq2* package which uses raw counts and models the normalization inside the negative binomial model.

## RESULTS
### Survival-based gene set enrichment analysis

In this study, we applied the SGSEA package and Shiny app to analyze the KIRC (Kidney Renal Clear Cell Carcinoma) dataset. KIRC is one of the most common and aggressive forms of kidney cancer, with a high mortality rate of around 22% has died from this kind of cancer for people who are diagnosed with kidney cancer, and is more common in men. According to recent studies, the 5-year survival rate for KIRC patients varies widely depending on the stage at diagnosis, with overall survival rates ranging from 70% to 90% for localized stages, but dropping significantly for metastatic cases (*National Cancer Institute, 2023*; *Hu, Zeng & Liu, 2019*). This highlights the need for effective tools that can identify the biological pathways associated with KIRC survival and mortality. As an example output of the *SGSEA* package and Shiny app, Table 1 shows the pathway analysis statistics using KIRC dataset. A positive normalized enrichment score (NES) indicates the pathway is enriched with genes having larger log hazard ratios and a larger log hazard ratio for a certain gene means that KIRC patients who have an increasing expression of

**Table 1  SGSEA package and Shiny app pathway analysis statistics using KIRC data.** Example output from the SGSEA package and Shiny app showing pathway analysis statistics based on kidney cancer data.

| Pathway | pval | padj | Log2err | ES | NES | Size | leadingEdge |
|---|---|---|---|---|---|---|---|
| Homo sapiens: 2-LTR circle formation | $7.9734 \times 10^{-1}$ | $8.9529 \times 10^{-1}$ | 0.0475 | 0.3411 | 0.7396 | 7 | BANF1, H… |
| Homo sapiens: A tetrasaccharide linker sequence is r… | $2.1229 \times 10^{-1}$ | $4.4949 \times 10^{-1}$ | 0.1110 | 0.4041 | 1.2085 | 26 | B3GAT3, … |
| Homo sapiens: ABC transporter disorders | $3.5335 \times 10^{-8}$ | $7.6478 \times 10^{-7}$ | 0.7195 | 0.5852 | 2.1875 | 77 | PSMD3, P… |
| Homo sapiens: ABC transporters in lipid homeostasis | $4.3452 \times 10^{-1}$ | $6.5931 \times 10^{-1}$ | 0.1106 | −0.3203 | −1.0100 | 18 | PEX19, A… |
| Homo sapiens: ABC family proteins mediated transport | $6.5682 \times 10^{-7}$ | $1.0755 \times 10^{-5}$ | 0.6594 | 0.2518 | 2.0369 | 102 | PSMD3, P… |
| Homo sapiens: ADORA2B mediated anti-inflammator… | $9.8930 \times 10^{-1}$ | 1.0000 | 0.0959 | −0.1570 | −0.7445 | 121 | GNB1, GN… |
| Homo sapiens: ADP signalling through P2Y purinocep… | $1.4815 \times 10^{-1}$ | $3.6462 \times 10^{-1}$ | 0.2043 | −0.3815 | −1.2984 | 25 | GNB1, GN… |
| Homo sapiens: ADP signalling through P2Y purinocep… | $2.7907 \times 10^{-1}$ | $5.2392 \times 10^{-1}$ | 0.1511 | −0.3487 | −1.1239 | 21 | GNB1, GN… |
| Homo sapiens: AKT phosphorylates targets in the cyt… | $2.0093 \times 10^{-1}$ | $4.3438 \times 10^{-1}$ | 0.1183 | 0.4739 | 1.2348 | 14 | CASP9, A… |
| Homo sapiens: AKT phosphorylates targets in the nuc… | $2.7632 \times 10^{-1}$ | $5.2070 \times 10^{-1}$ | 0.1336 | −0.4553 | −1.1478 | 9 | FOXO4, F… |

this gene tend to have a higher risk of dying. Conversely, a negative NES represents the pathway is enriched with genes having smaller log hazard ratio, linked to better survival. We consider a pathway significantly enriched if the adjusted *p*-value is below 0.15, as this threshold is commonly used in omics data to capture biologically meaningful pathways while maintaining better FDR control.

Numerous studies have suggested that proliferation-related biological functions are significantly associated with KIRC mortality, while pathways involved in metabolism and oncogenic transformation may be linked to better survival outcomes (*Fritz & Fajas, 2010*; *Cui et al., 2020*). For example, the mitotic cell cycle pathway, which is involved in cellular proliferation, has been found to be significantly up-regulated in aggressive KIRC tumors and associated with poor prognosis by *Golias, Charalabopoulos & Charalabopoulos (2004)*, and our results, shown in, also identify this pathway is one of the most significant pathways with NES = 2.33, adjusted *p*-value = $1.5e^{-29}$, reinforcing its role in poor survival. Conversely, pathways associated with metabolism and oncogenic transformation, such as the RHO GTPase cycle (*Orgaz, Herraiz & Sanz-Moreno, 2014*), were enriched with genes that have smaller log hazard ratios, suggesting their potential role in improving survival outcomes for KIRC patients.

In summary, Fig. 2 displays the enrichment plot of top 10 significant pathways with positive NES, linked to higher mortality in KIRC patients, and the top 10 with negative NES (from the 11th row to the last row), linked to higher survival. Figure 3 displays the individual enrichment plot for a specific pathway of interest, *e.g.*, Mitotic Cell Cycle and RHO GTPase, produced by the *SGSEA* R package (which utilized for plot generation). In particular, The plot of RHO GTPase shows a higher density of lines on the right, indicating that the pathway is enriched with genes having smaller log hazard ratios (NES = −1.54, adjusted *p*-value = $2.5e^{-6}$). This pathway is implicated in oncogenic transformation, which has been well-documented in the literature for its involvement in tumor progression and cancer metastasis (*Joshi-Tope et al., 2005*; *Orgaz, Herraiz & Sanz-Moreno, 2014*).

To assess the control of the false discovery rate (FDR) under the null hypothesis, we performed 100 permutations on the original data. For each of the 2,541 human pathways

**Figure 2** *SGSEA package and Shiny app pathway analysis output using KIRC data: the top 10 significant pathways with positive NES and the top 10 significant pathways with negative NES.* Visual representation of the top pathways enriched for survival outcomes with positive and negative normalized enrichment scores (NES).

from the Reactome database, we calculated the estimated FDRs based on the proportion of times the pathway's adjusted *p*-value (padj) was less than 0.15. The average FDR across all pathways was 0.1552. This result confirms that the SGSEA method appropriately controls FDR within the expected range (alpha level = 0.15). Table 2 shows the FDR values for an example of 15 pathways.

## Tumor *vs.* normal gene set enrichment analysis

The result of the standard GSEA method using the *SGSEA* package based on the log fold change between tumor and tumor-adjacent normal tissue is shown in Fig. 4. The signaling pathway pertinent to immunoregulatory interactions between a lymphoid and a non-lymphoid cell is significantly enriched with up-regulated genes in tumor compared to normal (*Wang et al., 2022b*). On the contrary, cellular metabolism pathways (on the bottom) are the most significantly enriched pathways with down-regulated genes in tumor compared to normal (*Yang et al., 2014*).

## Comparison between case-only SGSEA and tumor *vs.* normal GSEA

Figure 5 presents a Venn diagram showing the overlapping significant pathways identified by case-only SGSEA and tumor *vs.* normal GSEA separately at a 15% significance level to keep in line with the threshold recommendation from the original GSEA paper (*Subramanian et al., 2005*). Out of the total 1,867 pathways, 139 significant pathways were found to be overlapping. Additionally, SGSEA identified 354 significant pathways that were not detected by GSEA.

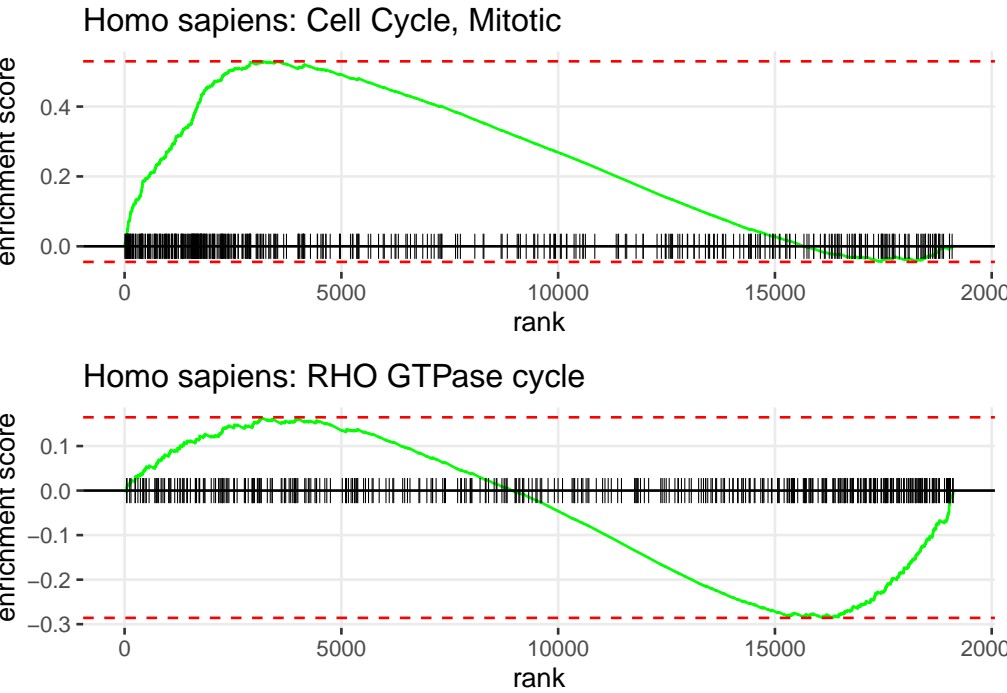

**Figure 3** *SGSEA* **package pathway analysis output using KIRC data: The individual enrichment plot of the top 1 significant pathway with positive NES (top) and negative NES (bottom).** Detailed enrichment plots for the most significant pathways linked to mortality and survival.

In Table 3, SGSEA identified the mitotic cell cycle and cell cycle checkpoints as two significant pathways enriched in genes with the largest log hazard ratios, while standard GSEA found these two pathways are statistically significantly (adjusted $p$-value < 0.15) enriched with up-regulated genes in tumor compared to normal. That is, these two pathways that are significantly associated with mortality were also found to be up-regulated in tumor *vs.* normal samples, suggesting that these two pathways are up-regulated with tumor *vs.* normal during tumor progression and will increase the risk of dying. On the other hand, SGSEA identified the RHO GTPase cycle and branched-chain amino acid catabolism as two pathways significantly (adjusted $p$-value < 0.15) associated with survival. However, typical GSEA found that the former one is not statistically significantly (adjusted $p$-value > 0.15) enriched in up-regulated genes with an NES score of 1.09, while the latter is significantly (adjusted $p$-value < 0.15) enriched in down-regulated genes with an NES score of −1.67, showing that the two methods can provide different insights.

## DISCUSSION

This work aims to demonstrate the utility of SGSEA in examining the association between disease survival and particular biological functions, while also providing tools to facilitate this analysis. We introduce the *SGSEA* R package and the accompanying Shiny app, demonstrating their functionality using a kidney cancer dataset as an illustrative example.

**Table 2   FDR estimation results based on 100 permutations.** FDR values are shown for an example of 15 pathways.

| Pathway | FDR |
| --- | --- |
| Homo sapiens: Hemostasis | 0.21 |
| Homo sapiens: Platelet degranulation | 0.05 |
| Homo sapiens: Immune System | 0 |
| Homo sapiens: Platelet activation, signaling and aggregation | 0.03 |
| Homo sapiens: Metabolism | 0.12 |
| Homo sapiens: Phase II - Conjugation of compounds | 0.07 |
| Homo sapiens: Acetylation | 0.03 |
| Homo sapiens: Biological oxidations | 0.11 |
| Homo sapiens: Drug ADME | 0.05 |
| Homo sapiens: Paracetamol ADME | 0.05 |
| Homo sapiens: Developmental Biology | 0 |
| Homo sapiens: Regulation of Insulin-like Growth Factor (IGF) transport and uptake by Insulin-like Growth Factor Binding Proteins (IGFBPs) | 0.05 |
| Homo sapiens: Metabolism of proteins | 0.02 |
| Homo sapiens: Post-translational protein modification | 0 |
| Homo sapiens: Post-translational protein phosphorylation | 0 |

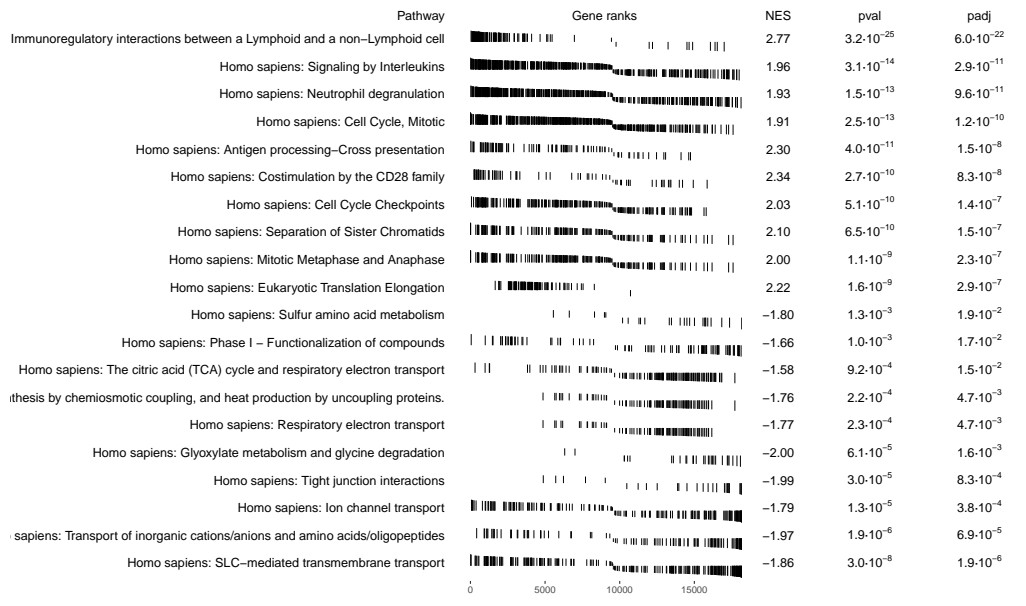

**Figure 4   GSEA pathway analysis using SGSEA package on KIRC data: top 10 significant pathways with positive NES and top 10 significant pathways with negative NES.** Comparison of GSEA results based on the traditional approach for pathways enrichment analysis.

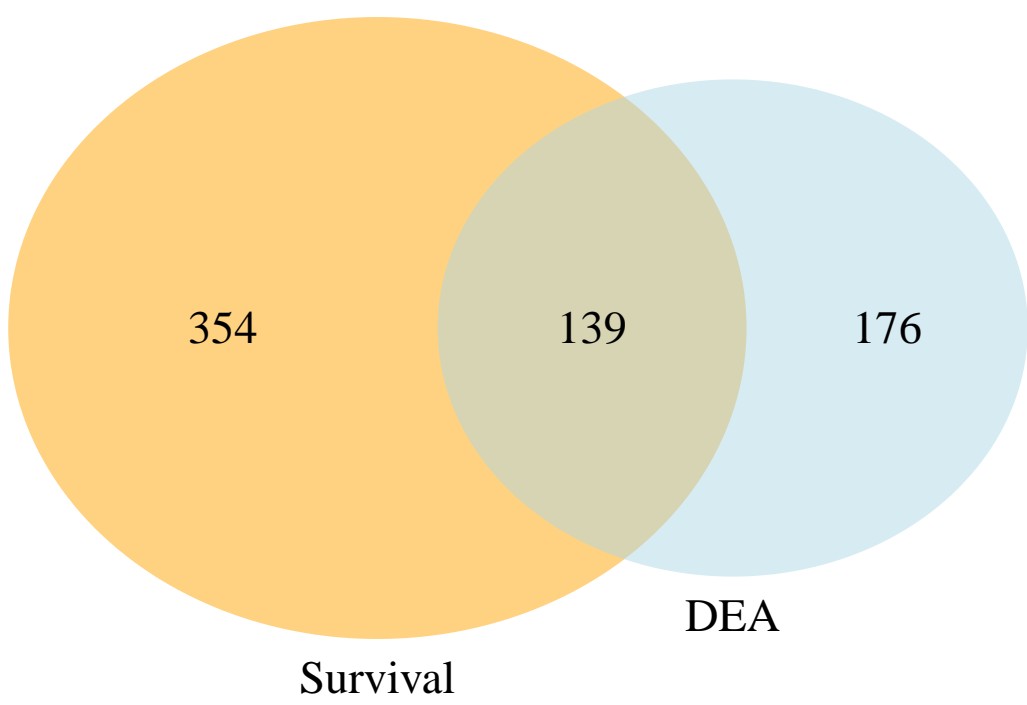

**Figure 5** **Venn diagram of comparison between SGSEA and GSEA.** The overlap between pathways identified by SGSEA and GSEA, highlighting the unique contributions of SGSEA in survival-related pathway analysis.

**Table 3** **Example result of pathways tested by SGSEA and GSEA.** The table shows the results of pathway enrichment analysis using SGSEA and GSEA, showing the significant pathways identified in both methods.

| | SGSEA | | GSEA | |
|---|---|---|---|---|
| | Adjusted *P*-value | NES | Adjusted *P*-value | NES |
| **Homo sapiens: Cell Cycle, Mitotic** | $8.9 \times 10^{-26*}$ | 2.36 | $1.2 \times 10^{-10*}$ | 1.91 |
| **Homo sapiens: Cell Cycle Checkpoints** | $1.8 \times 10^{-18*}$ | 2.43 | $1.4 \times 10^{-7*}$ | 2.03 |
| **Homo sapiens: RHO GTPase cycle** | $8.1 \times 10^{-6*}$ | −1.57 | $0.4 \times 10^{0}$ | 1.09 |
| **Homo sapiens: Branched-chain amino acid catabolism** | $1.6 \times 10^{-3*}$ | −2.13 | $0.9 \times 10^{-1*}$ | −1.67 |

**Notes.**
*Significant at 0.15.

Our Shiny app provides a user-friendly interface to conduct SGSEA efficiently. On the other hand, our R package provides greater flexibility for R programmers, allowing them to perform both SGSEA and GSEA within a single package and access cross-reference information.

Furthermore, the comparison between SGSEA and GSEA, illustrated in the Venn diagram, highlights that while SGSEA originates from GSEA, it offers unique insights and complementary information. While the original GSEA primarily focuses on assessing experimental differences, SGSEA incorporates survival information, making it particularly valuable for studying disease outcomes. This allows SGSEA to identify pathways linked to survival, which may not be detectable using traditional GSEA.

The data used for demonstration in this work are all cancer patients including censored cases. This highlights another key difference between SGSEA and a typical GSEA: SGSEA is case-only, and can handle censored data, whereas GSEA involves a comparison between treatment/case and control. Consequently, SGSEA specifically identifies pathways associated with disease survival within the context of abnormal tissues, such as up-regulated genes linked to mortality in KIRC patients. However, even when both types of data are available, as they are in these data, SGSEA identifies pathways associated with a disease's survival which might not be enriched using typical GSEA, such as pathway RHO GTPase cycle (*Liu et al., 2019*). That is because the drivers of survival differences do not necessarily need to be the processes that are changed by disease. If directly look at which pathway is up or down-regulated, it might miss these pathways that are associated with survival. Therefore, our finding suggests the potential utility of SGSEA in identifying key pathways involved in cancer progression and survival. For example, when comparing the significant pathways generated from the case-only SGSEA and those generated from tumor *vs.* normal GSEA, there are 139 (Fig. 5) overlapping pathways that are both related to disease outcome and disease state, conveying important biological information that might help researchers start their research more efficiently by focusing on the cell products or changes generated by these biological pathways first instead of those 176 (Fig. 5) significant pathways identified by typical GSEA. However, not all pathways associated with mortality are found to be up-regulated in tumor *vs.* normal tissues, nor are all pathways associated with survival found to be down-regulated in tumor *vs.* normal samples. Those 354 (Fig. 5) significant pathways from the Venn diagram, which do not differ between tumor and normal tissue but differ between survival and mortality, provide complementary information that helps researchers explore the biological processes underlying disease survival.

In summary, our study utilized the SGSEA R package and Shiny app to identify significant pathways associated with kidney cancer progression and survival, showcasing the effectiveness of SGSEA in this context. Pathways significantly associated with mortality (NES > 0 and adjusted *p*-value < 0.15) are enriched in genes whose expression is linked with higher mortality rates in KIRC patients. Similarly, pathways significantly associated with survival (NES < 0 and adjusted *p*-value < 0.15) are enriched in genes whose expression is linked with higher survival rates in KIRC patients. That is, the risk of dying will increase for KIRC patients who have increasing gene expression in pathways that are statistically significantly associated with mortality or decreasing gene expressions in pathways that are significantly associated with survival. Specifically, the pathways most closely linked to mortality are related to proliferation. This result has been validated in previous studies demonstrating that proliferation is associated with mortality in KIRC (*Morgan et al., 2018*; *Xiao et al., 2018*; *Wang et al., 2023*; *Zhang et al., 2016*).

However, it is important to note that there are certain constraints associated with this approach. These limitations primarily stem from the dependence on pre-existing pathway databases and the utilization of the GSEA method, which assesses coordinated change in gene expression. Also, currently only the Reactome database and Gene Ontology database is accessible for the *SGSEA* package. Another aspect to note is that we don't include the assumption checks for the Cox proportional hazards model in our tools. Our focus is

on obtaining coefficient values from thousands of Cox regression models rather than drawing inferences from individual models. As a result, it is not practical to perform assumption checks for thousands of models. While this approach may lead to potential biases in standard errors and *p*-values for some Cox models with invalid proportional hazard assumptions, our analysis does not rely on inferences from these individual models. Instead, we test the distribution of the ranks of coefficients using a permutation approach, in order to identify biological functions associated with mortality and survival. Using this approach, researchers may more accurately be able to determine the key drivers of the processes involved, which will not always contain the genes with the highest hazard ratios. This highlights the insufficiency of solely relying on survival analysis methods like Cox proportional hazards models to investigate the association between transcriptomic variations and clinical outcomes. In summary, our study confirms that pathways associated with a disease outcome might not always align with a disease state. Our *SGSEA* package and the Shiny app can help researchers conduct such functional enrichment analyses and gain insights into the complex relationship between transcriptomic variations and clinical outcomes.

## ACKNOWLEDGEMENTS

We would like to express our appreciation to Devin Koestler, Dong Pei, Rachel Griffard-Smith, Whitney Shae, Emily Schueddig for their invaluable feedback and support.

### Funding

This work was supported by the National Cancer Institute (NCI) Cancer Center Support Grant P30 CA168524, the Kansas Institute of Precision Medicine COBRE P20 GM130423, and a CTSA grant from NCATS awarded to the University of Kansas for Frontiers: University of Kansas Clinical and Translational Science Institute (# UL1TR002366). The contents are solely the responsibility of the authors and do not necessarily represent the official views of the NIH or NCATS. The funders had no role in study design, data collection and analysis, decision to publish, or preparation of the manuscript.

### Grant Disclosures

The following grant information was disclosed by the authors:
The National Cancer Institute (NCI) Cancer Center: P30 CA168524.
The Kansas Institute of Precision Medicine COBRE:  P20 GM130423.
A CTSA grant from NCATS awarded to the University of Kansas for Frontiers.
University of Kansas Clinical and Translational Science Institute: # UL1TR002366.

### Competing Interests

The authors declare there are no competing interests.

## Author Contributions

- Xiaoxu Deng conceived and designed the experiments, performed the experiments, analyzed the data, prepared figures and/or tables, authored or reviewed drafts of the article, and approved the final draft.
- Jeffrey Thompson conceived and designed the experiments, authored or reviewed drafts of the article, and approved the final draft.

## Data Availability

The Kidney Renal Clear Cell Carcinoma dataset (KIRC, version 2016_01_28), part of The Cancer Genome Atlas (TCGA) project, is available at The Broad Institute's Genome Data Analysis Center (GDAC) Firehose website: https://gdac.broadinstitute.org.

The SGSEA R package is available at GitHub and Zenodo:

- https://github.com/ShellsheDeng/SGSEA.

- ShellsheDeng. (2025). ShellsheDeng/SGSEA: First release of SGSEA (v1.0.0). Zenodo. https://doi.org/10.5281/zenodo.15099143

The Shiny app is available at: https://biostats-shinyr.kumc.edu/SGSEA/

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
