# Peer review of "An R package for survival-based gene set enrichment analysis"

_PeerJ, doi:10.7717/peerj.19489_

## Round 0.1 · original submission · Major Revisions

The reviewers seem to appreciate the technique and tool you have proposed. One reviewer asked for additional data and to make it easier to use the tool. Two reviewers asked for improvements to the manuscript itself. Please consider these requests carefully and submit a revised version.

Reviewer 1 ·

Basic reporting

The manuscript introduces the SGSEA method, which adapts Gene Set Enrichment Analysis (GSEA, specifically fgsea) by replacing traditional statistics with hazard ratios derived from Cox proportional hazards models. This adaptation allows for the direct integration of survival data into enrichment analyses. The authors have unified the calculation of hazard ratios and the fgsea analysis into a single R package and have developed a Shiny app to enhance user accessibility.

Experimental design

Testing the Null Hypothesis: The authors should conduct simulations to demonstrate that the method maintains the false discovery rate (FDR) at the specified alpha level under the null hypothesis. This can be achieved by randomizing the initial data numerous times (for example, performing hundreds of permutations) and assessing whether the FDR is controlled appropriately. This validation is critical for establishing the method's reliability and may uncover potential issues, including biases or errors in data manipulation.

Including Additional Ontologies/Annotation Systems: While Reactome pathways are valuable, including other annotation systems, especially the Biological Process category, which is one of the most widely used annotation systems in genomics, is crucial and makes the package more appealing. Incorporating GO terms would greatly enhance the utility of the SGSEA package.

Validity of the findings

Overall, the manuscript is clear and well-written, addressing an important gap in functional enrichment analysis by linking gene expression directly with survival outcomes. The manuscript could be substantially strengthened by including additional data:

1. Downloadable Example Data: The Shiny app and the R package should include downloadable example datasets and scripts that users can run directly. This improves user experience and facilitates easier adoption of the tool.

2. Export of Ranked Gene Lists for Downstream Analysis: The package should offer an option to export the ranked genes along with their Cox hazard ratios as a list or data frame. This flexibility enables users to validate and compare results across different platforms.

3. Customization of Cox Model Parameters: The SGSEA package should allow users to customize the Cox proportional hazards model by including additional covariates. This is important for enhancing the applicability of the method and improving the accuracy of the survival analysis results.

Additional comments

Absence of Support in the Shiny App: The app should include a help link or a support email address to assist users with any technical issues or inquiries.

Licensing Information: The authors have attached the GPLv3 license to the code but should clarify the licensing of the software tool and the results, including which license pertains to the usage of the Shiny app. All the relevant licensing terms should be included both in the manuscript and in the Shiny app.

·

Basic reporting

NO comment

Experimental design

no comment

Validity of the findings

no comments

Reviewer 3 ·

Basic reporting

1. While most of the text can be understood well, there are a few sentences that can benefit from being reworded for clarity, particularly this one: "Furthermore, the survival differences among those with the disease are not what drove the results of the GSEA analysis or survival analysis tests only among the treatment differences rather than case-only, which creates a blind spot in the results."
2. The figures are not referenced correctly, for example: "Figure 5 presents a Venn diagram showing the overlapping significant pathways." Fig 4 is the Venn diagram
3. The case study section could use more citations, there are some in the discussion section but it's important to add these too to the results/use case to highlight the validity of the results of the app. Examples:
3.1 "indicates that proliferation-related biological functions are associated with KIRC mortality, while metabolism and oncogenic transformation pathways are associated with KIRC survival, for people who have increased expression."
3.2 "the pathway on the bottom with negative NES is the RHO GTPase cycle which is significantly enriched in genes with the smallest log hazard ratios (NES=-1.57, adjusted p-value=8.1ΓΏ 2 6). The biological function of this pathway is related to oncogenic transformation 11." You cited reactome here but I don't think this is sufficient.

4. The case study could be reworked to be more of a case study and not just a demo of the app. First, why choose KIRC? What's the mortality rate of this disease? What is known about what pathways or genes that affect the mortality of KIRC? These should then be reflected in the results of the enrichment analysis with relevant citations.

5. Please put the GitHub and website in the abstract and/or at a Code Availability section

Experimental design

The methods look novel enough but the case study needs a bit of rework.

Validity of the findings

The authors created a novel method that combined GSEA and survival analysis to find enriched pathways that can affect survival/mortality. The data and the version they used are listed which helps in replicating the study. They also listed all of the R libraries they used.

Additional comments

It would be better if you had a button that loads and/or downloads an example on your website

---

## Round 0.2 · accepted · Accept

Thank you for addressing the reviewers' comments. I have assessed the revised changes and found that they have addressed the reviewers' comments. The manuscript is now ready for publication.